# Thermal Susceptibility of Nickel in the Manufacture of Softeners

Pino P. Álvarez-Castellanos [1], Pablo Fernández-Arias [2], Diego Vergara [2,*] and Francisco J. San-José [1]

1   Analysis and Sustainability Production of Agro-Industrial Sector Research Group, Catholic University of Ávila, C/Canteros, s/n, 05005 Ávila, Spain
2   Technology, Instruction and Design in Engineering and Education Research Group, Catholic University of Ávila, C/Canteros, s/n, 05005 Ávila, Spain
*   Correspondence: diego.vergara@ucavila.es

**Abstract:** The chemical industry includes a wide range of factories focused on obtaining final products as: (i) plastics; (ii) chemical fibers; (iii) rubber; (iv) perfumery and cosmetic products; and (v) cleaning products. Although the level of safety in the activities and installations of this sector is very high, the use of dangerous substances implies an increased risk of suffering an accident involving the emission of hazardous substances, as well as endangering the safety of workers. In the case of the manufacture of softeners, the presence of isopropanol ($C_3H_8O$), and dimethyl sulfate ($(CH_3)_2SO_4$), have been reported to be the accident cause in most of the cases. The European accident database (eMars) reported an accident in which the presence of impurities of nickel (Ni) in the hydrogenated tallow used as raw material for softener production may have increased thermal reactivity and the chances of spontaneous combustion. This paper analyzes the results obtained with the Maciejasz Index (MI) to understand the thermal susceptibility of these substances in liquid state. The results show that combinations of nickel (hydrogenated tallow catalyst) with other liquid substances (isopropanol, dimethyl sulfate, and sulfuric acid) are not sufficiently reactive with oxygen to cause a spontaneous combustion.

**Keywords:** chemical industry; esterquat softener; thermal susceptibility; Maciejasz Index (MI); spontaneous combustion





## 1. Introduction

The main target of any industry is to obtain repair, maintenance, transformation, or reuse of industrial products, as well as the use, recovery, and disposal of waste or by-products. To achieve this goal, it is necessary to establish a business strategy based on two fundamental pillars: (i) the sustainability and efficiency of the system; and (ii) the safety of the installation and of the workers who carry out their professional activity [1,2]. However, numerous events endanger these pillars of the business strategy of the industrial sector, causing considerable material damage and sometimes even countless fatalities, unfortunately.

The International Labour Organization (ILO)—a specialized agency of the United Nations (UN) that has brought together governments, employers, and workers from 187 UN Member States—estimates that every year some 2.3 million women and men succumb to work-related accidents or diseases. (Approximately 6000 deaths per day). Worldwide, it is estimated that some 340 million occupational accidents and 160 million victims of work-related illness occur annually [3]. Combining the results of injuries and diseases, at the beginning of the 21st century the annual number of deaths of workers resulting from occupational exposure was about two million, with 350,000 deaths from injuries and about 1.65 million deaths from work-related diseases [4,5].

One of the industrial sectors that has most promoted the safety of its facilities and its workers has been the chemical sector [6,7]. The aim of this transversal and integrating

industrial sector is extracting and processing raw materials, natural and synthetic, and their transformation into new substances and components. On the other hand, the chemical industry divides into two major subsectors: (i) the basic chemical industry: which uses raw materials or resources in their natural state, which it transforms and converts into intermediate products; and (ii) the processing chemical industry: which uses as raw materials the intermediate products produced by the basic chemical industry. Within this last subsector, various industrial activities focused on obtaining different final products are included: (i) plastics; (ii) chemical fibers; (iii) rubber; (iv) perfumery and cosmetic products; (v) cleaning products [8–12]; and (vi) resin materials [13].

Although the level of safety in chemical installations is very high, the use of hazardous substances poses a high risk of suffering a serious event, which could involve the emission or discharge of dangerous substances, explosion risk [14,15], as well as a situation of risk to the safety of workers [16–19]. An example of this high risk in chemical installations is the accident that originated in 1976 in a small chemical plant in the Italian municipality of Seveso [20,21]. The accident produced the release to the environment of high amounts of Tetrachlorodibenzo-p-dioxin (TCDD dioxin) [22,23], which reached several inhabited areas, damaging the health of many of its inhabitants and the environment in between. Following the accident, the European Economic Community agreed in 1982 on new safety rules for industrial plants using dangerous chemical compounds through the internationally known European Directive 82/501/EEC or "Seveso Directive" and its subsequent amendments [24,25].

Within the chemical sector, household softeners are manufactured using intermediates produced in the basic chemical industry [26–30]. Accidents in these types of industries (Table 1), according to the record found in the eMARS accident database [31]—which represents the European Union major accident reporting system—are mainly due to the use of different chemicals [32]: (i) Isopropanol; (ii) Dimethyl sulfate; (iii) Amines participating in esterquat; and (iv) the resulting product itself.

**Table 1.** Chemical accidents in Europe related to Isopropanol and esterquats.

| Start Date | Accident Title | Substances Involved | Causes Description |
|---|---|---|---|
| 8 Jun. 1988 | Explosion followed by a fire caused by wrong mixing operation in a reactor [33]. | Sodium Hypochlorite. | Wrong mixing in a reactor, more than 100 substances involved including, isopropanol. |
| 1 Oct. 2000 | Tonnes of mixed chemical wastes consumed in the fire of a site for treatment and storage of chemical waste [34]. | Isopropanol. | Fire spreads to a large number of flammable substances. Start of fire: Isopropanol. |
| 11 Feb. 2005 | Plant destroyed by the fire of a broken oil cooling system caused by overpressure due to runaway reaction [35]. | Sodium Ethylate. | The sodium borohydride in a mixture with alcohols caused an exothermic reaction, inflammation, and the flame broke the oil cooling system. |
| 26 Jul. 2006 | Large burning of hydrocarbon chemicals in a terminal [36]. | Acetone. Styrene. Toluene. Methanol. Isopropanol. Hexane. | It may be a combination of a leakage (even not from xylene) and static electricity or spark caused by accidental strike/friction of metal equipment. It is under juridical investigation. |
| 24 Mar. 2007 | Spillage of rinsing water containing chromium trioxide in a galvanic installation [37]. | Sodium bisulfite. | The event was triggered by the opening of the threaded pipe joint (on the pressure side) connecting the shutoff ball valve with the circulating pump. The rinsing liquid (under pressure) sprayed beyond the catching cup. |
| 27 May. 2011 | Release of substances and subsequent fire at a surfactant production plant [38]. | Phthalic anhydride. | The ignition of the phthalic anhydride was probably caused by sparking resulting from short-circuiting of electrical cables positioned close to the ground. The cables were flooded by the hot phthalic anhydride, melting their insulation. |
| 27 Aug. 2015 | Deflagration reactor accident during manufacturing process softening Roquat 75 [39]. | Isopropyl alcohol. Diethylenetriamine. Sodium bromide. Hydrogen peroxide. Dimethyl sulfate. | The SM 75 ROQUAT finished product is a flammable solid, skin irritant and presented specific toxicity effects of drowsiness and dizziness. |

In the case of the accident caused by the deflagration of the reactor during the manufacturing process of the Roquat 75 softener, in addition to the substances mentioned above: (i) isopropanol; (ii) dimethyl sulfate; (iii) amines participating in esterquat; and (iv) the resulting product itself; it is necessary to add the presence of traces of nickel [39] as a differentiating element. Given this situation, if proven that traces of nickel and other chemical compounds were the initiating elements of the event, the procedures and regulations of surfactant companies should be reconsidered and modified, including traces of nickel as potential risk when the fat has been hydrogenated.

For these reasons, this paper aims to analyze the reactivity of impurities of the catalyst (nickel) with the rest of the substances that make up the manufactured esterquat and to check whether this reactivity can cause an explosion. Thus, it is necessary to resort to the Maciejasz Index (MI) test, used to characterize thermal [40] susceptibility or predisposition to oxidation [40–42].

## 2. Materials and Methods

To achieve the research aims, it is crucial to develop a structured methodology in different phases (Figure 1): Phase I: fabrication process of fabric softeners and importance of nickel; Phase II: test for the reactivity of catalyst Ni in the batch of hydrogenated tallow; Phase III: analysis of the results obtained.

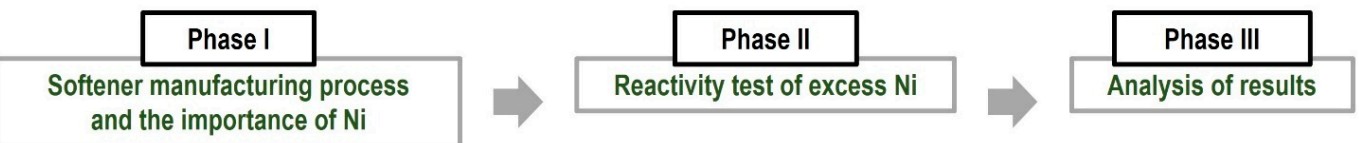

**Figure 1.** Methodology process.

### 2.1. The Production Process of Fabric Softeners Using Nickel and Esterquats

Softeners contain fatty acid esterquats as main components. For the formation of esterquats (Figure 2) a tertiary alkanolamine is esterified with a mixture of fatty acids or their esters, such as glycerides; after this step, there is a quaternization reaction of the alkanolamine esterified with an alkylating agent [43]. If compared with other fats on the market, with higher industrial value, hydrogenated tallow is a common raw material in this industry, due to its low price. Hydrogenation of oils and fats aims to raise their melting point and gain more stability. The hydrogenation reaction (Figure 2) adds hydrogen atoms to tallow until the total saturation of its double bonds is achieved. For this exothermic and irreversible reaction to happen, pressurized hydrogen, temperatures above 156 °C, and nickel as a catalyst are needed.

In industrial softener formulations, it is common practice to use dimethyl sulfate as the alkyl agent and Isopropanol as its solvent. It is known that Isopropanol stays unreactive during the hold processes and remains in the final product. After obtaining the softener, the last step is bleaching the product with a solution of $H_2O_2$ (30% in water).

The manufacturing process of hydrogenated fats (Figure 3) at the industrial level is a batch process carried out in 2 jacketed and agitated tank reactors, which controls reaction time, pressure and working temperature, and the hydrogen addition. The mix of the catalyst (nickel source) with the oil happens before entering the reactor. At the exit of the reactor, the hydrogenated sebum passes through a filter plate which retains the catalyst for later reuse.

Hydrogenated tallow is one of the initial components in the manufacture of softeners [44]. The sebum is a lipid matrix of animal origin whose main component is triglycerides of fatty acids with different degrees of unsaturation [45].

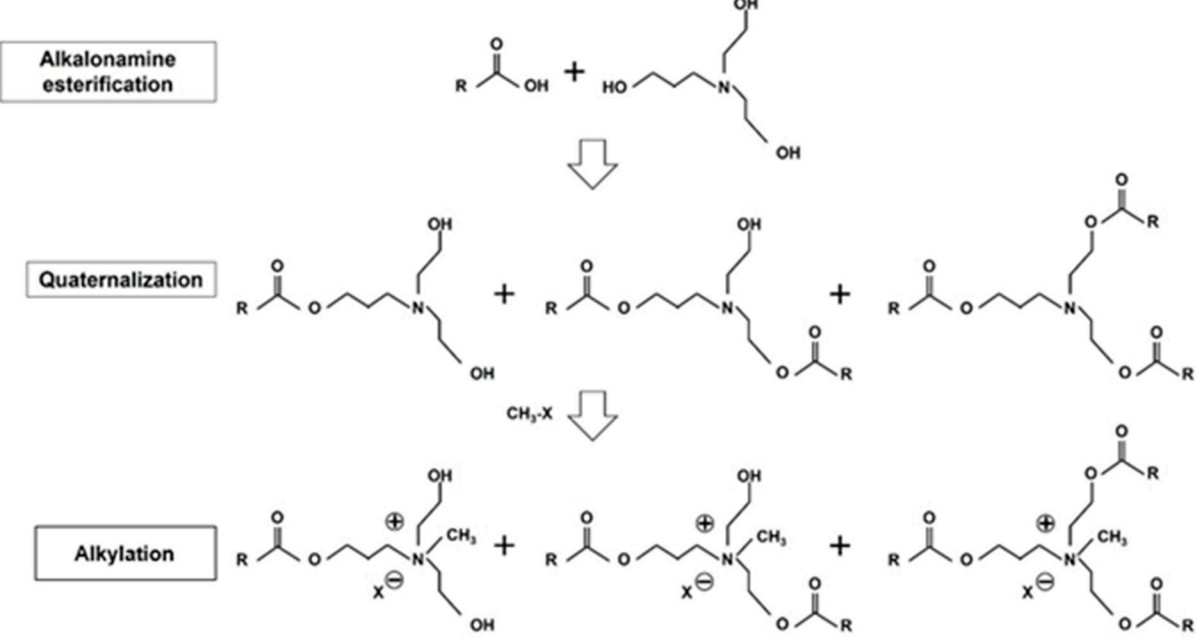

**Figure 2.** Production process of fabric softeners.

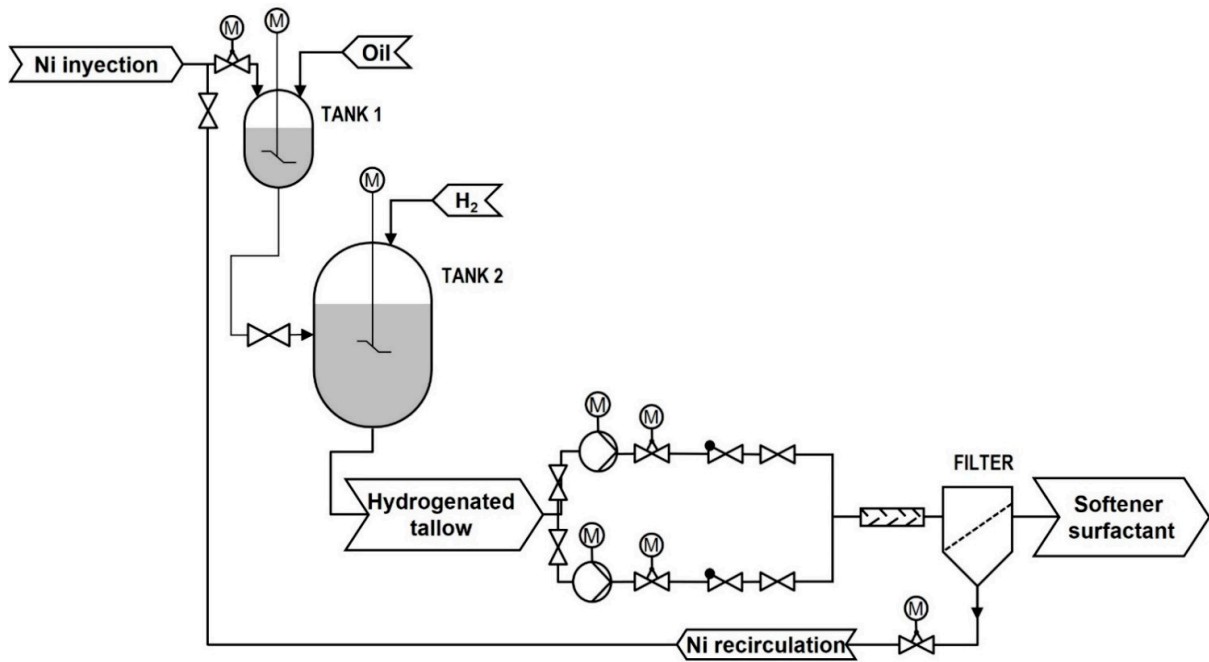

**Figure 3.** Experiment for determining the MI.

Although some nickel can come out with sebum as an impurity, it is possible to limit its concentration to values below 25 ppm and thus use it as a feed product [46,47]. However, surfactant factories generally do not set any limits on the concentration of nickel. Also, it may be assumed that the lack of complaints from hydrogenated tallow customers indicates an adequate quality standard. The possibility that a certain amount of Nickel could increase explosion risk would completely modify the safety standards required for the raw material in the sector. Thus, the study presented in this paper is of great interest in the surfactant manufacturing sector.

Nickel loses reaction capacity as it oxidizes in each cycle, going from pure nickel to nickel (II) ion. Consequently, after recovering the catalyst, some of it is rejected and replaced with fresh inputs. So, the nickel found in the hydrogenated tallow is more oxidized than the

fresh. It is less active and has less reaction power with hydrogen peroxide ($H_2O_2$) because it is partially or completely oxidized.

### 2.2. Reactivity Test for Catalyst Traces of Ni as an Impurity in Hydrogenated Tallow

This paper tests whether a high-range concentration (200 ppm) of traces of nickel could influence the reactants or softener components with the $H_2O_2$ used in the bleaching reactions, causing a high-temperature oxidation reaction that eventually might cause an explosion. Due to the use of hazardous compounds within the ATEX regulations of explosive atmospheres, the experimental part was carried out in a reference laboratory in Spain [48–51], *Laboratorio Oficial José Maradiaga* (LOM), which is the only Spanish laboratory certified by the European Union to work in hazardous areas [52].

The test used to determine the reactivity of traces of nickel in hydrogenated tallow in the softener bleaching process is called Maciejasz Index (MI), used to characterize thermal susceptibility or predisposition to oxidation [40–42]. It was necessary to perform nickel reactivity tests with different compounds present in the manufactured esterquat to determine the influence of nickel. As shown in Figure 4, the method lets a certain amount of product react with $H_2O_2$. The operative procedure (Table 2) consists of putting 10 g of the substance in a Dewar glass, adding 30 mL of 30% $H_2O_2$. To check whether a delayed reaction may occur when the temperature is not reached, time is postponed at least 20 min. At least six replicates were run. The Maciejasz Index (MI) is calculated with the measured time as follows: MI = 100/t [40].

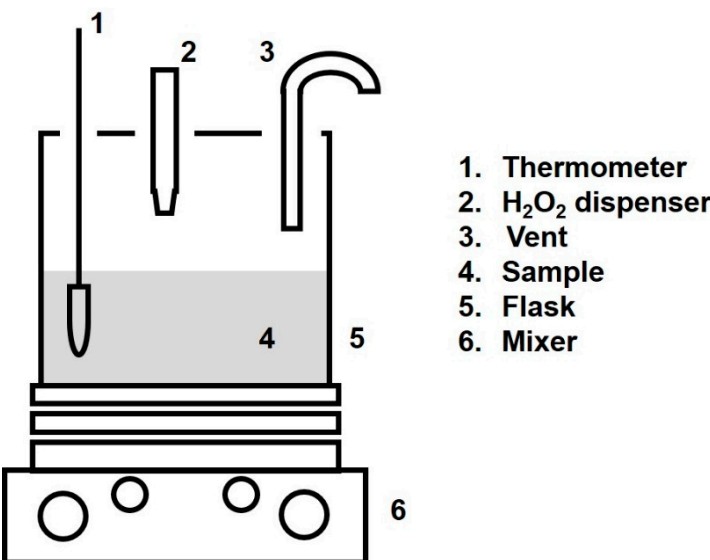

1. Thermometer
2. $H_2O_2$ dispenser
3. Vent
4. Sample
5. Flask
6. Mixer

**Figure 4.** Experiment for determining the MI.

**Table 2.** Thermogravimetry Maciejasz Index (MI) variables measured.

| Variable | Value |
|---|---|
| Volume | 150 mL |
| Initial temperature | 15–18 °C |
| Final temperature | 80 °C |
| Ambient temperature | 25 °C |
| Time test | 20 min |
| Heating rate | 3 °C/min |
| Maciejasz Index (MI) | 100/t |

These variables, shown in Table 2, test the tendency towards combustion or reactivity to a $H_2O_2$-like product. In each replicate, 10 g product samples were mixed with 30 mL $H_2O_2$ (30%) and allowed to react. The temperature measured just before adding $H_2O_2$ is

the initial temperature. The final time is when the temperature stabilizes. If the temperature reaches 80 °C, the time is measured, and the test is over. The Maciejasz Index MI is a method applied in the mining sector. As the esterquat undergoes a bleaching process with hydrogen peroxide ($H_2O_2$) during the industrial process, the accident report considered that their reactivity with residual nickel detected in the tallow was a plausible cause of the incident. It is known that if the Maciejasz Index (MI) is greater than 10, the substance is sufficiently reactive with oxygen to consider that spontaneous combustion may occur.

In contrast to other papers analyzing the synergetic behavior of mixtures in the gaseous state [53], this study aims to test whether the existence of liquid nickel in sebum would catalyze the decomposition of $H_2O_2$, causing an increase in temperature until reaching the deflagration temperature of isopropanol. The substances that can generate an explosion depending on the formulation of this liquid product are: (i) isopropanol ($C_3H_8O$); (ii) dimethyl sulfate (($CH_3$)$_2SO_4$), a methylation reagent that could cause the reaction either through an excessive dosage of dimethyl sulfate or its traces from an uncompleted reaction; and (iii) sulfuric acid ($H_2SO_4$), which is obtained during the methylation reaction itself by the breakdown of dimethyl sulfate. The combination of isopropanol with other liquid substances, including nickel, limits the heating range, as the evaporation of isopropanol occurs from 80 °C onwards [54]. Therefore, it is considered that there will always be isopropanol and possibly some traces of the other products (2%). It has been reported that Isopropanol and dimethyl sulfate caused accidents already; it is relevant to consider them as the source of the explosion [39,40]. To detect the possible effect of this catalyst, 200 ppm of nickel was added to some of these samples. Table 3 shows the concentrations of these three substances in the different samples tested.

**Table 3.** Samples tested.

| Sample | Description |
|--------|-------------|
| S1 | $C_3H_8O$ |
| S2 | $C_3H_8O$ + 200 ppm Ni |
| S3 | $C_3H_8O$ + 2% ($CH_3$)$_2SO_4$ |
| S4 | $C_3H_8O$ + 2% ($CH_3$)$_2SO_4$ + 200 ppm Ni |
| S5 | $C_3H_8O$ + 2% $H_2SO_4$ |
| S6 | $C_3H_8O$ + 2% $H_2SO_4$ + 200 ppm Ni |

## 3. Results and Discussion

Below (Table 4), the results obtained in the different samples are shown: S1: $C_3H_8O$; S2: $C_3H_8O$ + 200 ppm Ni; S3: $C_3H_8O$ + 2% ($CH_3$)$_2SO_4$; S4: $C_3H_8O$ + 2% ($CH_3$)$_2SO_4$ + 200 ppm Ni; S5: $C_3H_8O$ + 2% $H_2SO_4$; S6: $C_3H_8O$ + 2% $H_2SO_4$ + 200 ppm Ni. These samples are ordered to observe the influence of the components in the formulation, with or without nickel. It can be observed that there is no significant difference between the reagent with Ni (samples S2, S4 and S6) and without Ni (samples S1, S3 and S5). In none of these mixtures with or without Ni is the temperature significantly different (Table 5). As for the temperature in the $H_2O_2$ mixture, it can be observed that there is no significant difference between the reagent with Ni (samples S2, S4 and S6) and without Ni (samples S1, S3 and S5).

Given the results obtained, it is possible to affirm that none of the products or mixtures considered is highly reactive to hydrogen peroxide, at 30%. In addition, it shows that the catalyst (nickel) did not have any significant influence in this reactivity. The deflagration during the manufacturing process of the softener Roquat 75 [39] could have any other cause than nickel traces, since it is proven that the reactivity to hydrogen peroxide of the whitening products used during the production has no significant differences with or without nickel.

Therefore, the results indicate that despite the doubts raised in the accident report [39] regarding the possible influence of nickel on the explosion, there is no evidence that Ni is one of the causes that can cause this type of explosion in the surfactant sector. In this way,

this study reveals that it is not necessary to rethink the current regulations in this sector regarding the limitation of ppm of Ni in the manufacturing process of esterquat softeners.

**Table 4.** Results sample S1 ($C_3H_8O$).

| Sample | Test | Initial Temperature (°C) | Final Temperature (°C) | Time Test (min) | Maciejasz Index (MI) |
|---|---|---|---|---|---|
| S1 | S1-1 | 20 | 24 | 22 | 4.54 |
| | S1-2 | 17 | 22 | 23 | 4.34 |
| | S1-3 | 17 | 22 | 20 | 5.00 |
| | S1-4 | 18 | 23 | 23 | 4.34 |
| | S1-5 | 18 | 23 | 22 | 4.54 |
| | S1-6 | 19 | 24 | 25 | 4.00 |
| S2 | S2-1 | 20 | 23 | 43 | 2.32 |
| | S2-2 | 20 | 25 | 20 | 5.00 |
| | S2-3 | 21 | 26 | 20 | 5.00 |
| | S2-4 | 21 | 26 | 26 | 3.84 |
| | S2-5 | 22 | 27 | 21 | 4.76 |
| | S2-6 | 23 | 27 | 22 | 4.54 |
| S3 | S3-1 | 21 | 26 | 20 | 5.00 |
| | S3-2 | 22 | 26 | 22 | 4.54 |
| | S3-3 | 22 | 26 | 20 | 5.00 |
| | S3-4 | 23 | 27 | 20 | 5.00 |
| | S3-5 | 23 | 27 | 20 | 5.00 |
| | S3-6 | 23 | 27 | 23 | 4.34 |
| S4 | S4-1 | 21 | 26 | 20 | 5.00 |
| | S4-2 | 18 | 22 | 22 | 4.54 |
| | S4-3 | 19 | 23 | 20 | 5.00 |
| | S4-4 | 20 | 25 | 21 | 5.00 |
| | S4-5 | 21 | 25 | 20 | 5.00 |
| | S4-6 | 21 | 26 | 26 | 4.34 |
| S5 | S5-1 | 23 | 25 | 30 | 3.33 |
| | S5-2 | 23 | 26 | 23 | 4.35 |
| | S5-3 | 24 | 27 | 20 | 5.00 |
| | S5-4 | 24 | 27 | 21 | 4.76 |
| | S5-5 | 25 | 28 | 20 | 5.00 |
| | S5-6 | 24 | 27 | 20 | 5.00 |
| S6 | S6-1 | 25 | 29 | 30 | 3.33 |
| | S6-2 | 25 | 27 | 20 | 5.00 |
| | S6-3 | 25 | 28 | 21 | 4.76 |
| | S6-4 | 25 | 28 | 21 | 4.76 |
| | S6-5 | 25 | 28 | 22 | 4.54 |
| | S6-6 | 25 | 27 | 20 | 5.00 |

**Table 5.** Global results samples tested.

| Sample | Description | ΔT (°C) | Maciejasz Index (MI) |
|---|---|---|---|
| S1 | $C_3H_8O$ | 5.0 | 4.46 |
| S2 | $C_3H_8O$ + 200 ppm Ni | 5.0 | 4.24 |
| S3 | $C_3H_8O$ + 2% $(CH_3)_2SO_4$ | 4.0 | 4.81 |
| S4 | $C_3H_8O$ + 2% $(CH_3)_2SO_4$ + 200 ppm Ni | 4.5 | 4.81 |
| S5 | $C_3H_8O$ + 2% $H_2SO_4$ | 3.0 | 4.57 |
| S6 | $C_3H_8O$ + 2% $H_2SO_4$ + 200 ppm Ni | 3.0 | 4.56 |

## 4. Conclusions

The chemical sector has been one of the industrial sectors that has most promoted the safety of its facilities and workers. However, numerous events throughout history show that, although the safety level in chemical activities and facilities is very high, the use of

hazardous substances poses a risk of suffering a dangerous event. Within the chemical processing industry, there are business activities aimed at obtaining household cleaning products from intermediates produced by the primary chemical industry, such as, for example, the production of household softeners. The present results conclude that even combined with other liquid substances such as isopropanol, dimethyl sulfate, and sulfuric acid, nickel (traces of catalyst used in hydrogenated tallow) is not sufficiently reactive with oxygen to consider that spontaneous combustion may occur.

**Author Contributions:** Conceptualization, P.P.Á.-C., P.F.-A. and D.V.; methodology, P.P.Á.-C., P.F.-A. and D.V.; formal analysis, P.P.Á.-C., P.F.-A. and D.V.; investigation, P.P.Á.-C.; data curation, P.F.-A. and D.V.; writing—original draft preparation, P.P.Á.-C., P.F.-A. and D.V.; writing—review and editing, P.P.Á.-C., P.F.-A., D.V. and F.J.S.-J.; funding acquisition, P.P.Á.-C. All authors have read and agreed to the published version of the manuscript.

**Funding:** This research was funding by the company Mateos S.L. (Spain).

**Conflicts of Interest:** The authors declare no conflict of interest.

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
