# Peer review of "Thermal Susceptibility of Nickel in the Manufacture of Softeners"

_processes, doi:10.3390/pr11030821_

Round 1

Reviewer 1 Report

The authors announce that the research examines the Maciejasz Index (MI) data to determine its thermal susceptibility. Findings shows that The results indicate that Nickel is not sufficiently reactive with oxygen to generate spontaneous combustion. The data is sufficient to prove the conclusion and mechanism. This article can be included in “Safety” after minor revisions.

1.     Is Table 2 really necessary?

2.     The heading 2.1 should be modified.

3.     The format of brackets, en dash, and hyphen did not follow the common usage of journal.

4.     A few grammatical and typo mistakes are still there in your manuscript. Please thoroughly check and revise the manuscript carefully.

5.     The certainty analysis for experimental results are missing.

6.     The content is plentiful, but some part of the reference literatures is kind of obsolete (in 3 years). Key publications should be cited as completed as possible. Please also clarify the novelty and application implication of your work in this section. I suggest authors refer to the latest literatures from “MDPI” and other safety journals. But please do not exceed 30% of all citations from MDPI. Authors may see the following reference while revising. 
https://doi.org/10.1007/s10973-022-11819-1;
https://doi.org/10.3390/pr10081581;
https://doi.org/10.3390/pr10081606;

Author Response

Please, find enclosed a detailed response.

Reviewer 2 Report

I would like to thank the authors for the well written work. I suggest some corrections, modifications and integration in the attached PDF. Please look at the column at the left page.

Author Response

(The authors gave the same response as above.)
